# Impacting the Remedial Potential of Nano Delivery-Based Flavonoids for Breast Cancer Treatment

**DOI:** 10.3390/molecules26175163

**Published:** 2021-08-26

**Authors:** Rakesh K. Sindhu, Rishu Verma, Twinkle Salgotra, Md. Habibur Rahman, Muddaser Shah, Rokeya Akter, Waheed Murad, Sidra Mubin, Parveen Bibi, Safaa Qusti, Eida M. Alshammari, Gaber El-Saber Batiha, Michał Tomczyk, Hayder M. Al-kuraishy

**Affiliations:** 1Chitkara College of Pharmacy, Chitkara University, Punjab 140401, India; verma.rishu2000@gmail.com (R.V.); vidhiattri2001@gmail.com (T.S.); 2Department of Pharmacy, Southeast University, Banani, Dhaka 1213, Bangladesh; 3Department of Global Medical Science, Wonju College of Medicine, Yonsei University, Gangwon, Wonju 26426, Korea; rokeyahabib94@gmail.com; 4Department of Botany, Abdul Wali Khan University Mardan, Mardan 23200, Pakistan; waheedmurad@awkum.edu.pk (W.M.); parveentariq825@gmail.com (P.B.); 5Department of Botany, Hazara University Mansehra, Mansehra 21310, Pakistan; shahhu123@gmail.com; 6Biochemistry Department, Faculty of Science, King Abdul Aziz University, Jeddah 22230, Saudi Arabia; squsti@kau.edu.sa; 7Department of Chemistry, College of Sciences, University of Ha’il, Ha’il 55211, Saudi Arabia; eida.alshammari@uoh.edu.sa; 8Department of Pharmacology and Therapeutics, Faculty of Veterinary Medicine, Damanhour University, Damanhour 22511, Al Beheira, Egypt; gaberbatiha@gmail.com; 9Department of Pharmacognosy, Medical University of Białystok, ul. Mickiewicza 2a, 15-230 Białystok, Poland; michal.tomczyk@umb.edu.pl; 10Department of Clinical Pharmacology and Medicine, College of Medicine, Al Mustanysiriyia University, Baghdad 10011, Iraq; hayderm36@yahoo.com

**Keywords:** flavonoids, nanoparticles, breast cancer, drug delivery system, cancer therapeutics, anticancer, phytochemicals, apoptosis, epidemiological study

## Abstract

Breast cancer persists as a diffuse source of cancer despite persistent detection and treatment. Flavonoids, a type of polyphenol, appear to be a productive option in the treatment of breast cancer, because of their capacity to regulate the tumor related functions of class of compounds. Plant polyphenols are flavonoids that appear to exhibit properties which are beneficial for breast cancer therapy. Numerous epidemiologic studies have been performed on the dynamic effect of plant polyphenols in the prevention of breast cancer. There are also subclasses of flavonoids that have antioxidant and anticarcinogenic activity. These can regulate the scavenging activity of reactive oxygen species (ROS) which help in cell cycle arrest and suppress the uncontrolled division of cancer cells. Numerous studies have also been performed at the population level, one of which reported a connection between cancer risk and intake of dietary flavonoids. Breast cancer appears to show intertumoral heterogeneity with estrogen receptor positive and negative cells. This review describes breast cancer, its various factors, and the function of flavonoids in the prevention and treatment of breast cancer, namely, how flavonoids and their subtypes are used in treatment. This review proposes that cancer risk can be reduced, and that cancer can be even cured by improving dietary intake. A large number of studies also suggested that the intake of fruit and vegetables is associated with reduced breast cancer and paper also includes the role and the use of nanodelivery of flavonoids in the healing of breast cancer. In addition, the therapeutic potential of orally administered phyto-bioactive compounds (PBCs) is narrowed because of poor stability and oral bioavailability of compounds in the gastrointestinal tract (GIT), and solubility also affects bioavailability. In recent years, creative nanotechnology-based approaches have been advised to enhance the activity of PBCs. Nanotechnology also offers the potential to become aware of disease at earlier stages, such as the detection of hidden or unconcealed metastasis colonies in patients diagnosed with lung, colon, prostate, ovarian, and breast cancer. However, nanoformulation-related effects and safety must not be overlooked. This review gives a brief discussion of nanoformulations and the effect of nanotechnology on herbal drugs.

## 1. Introduction

Cancer is an inherited disease provoked by changes in the genetic code that regulate the functioning of cells, and is a crucial health complication over the world [1]. The incidence of cancer has increasing during the previous few years, but has decreased due to advancements in the treatment of cancer [2]. Epigenetic and genetic factors may cause the formation of tumor cells and their proliferation [3]. It is due to atypical growth of cells and has the potential to interfere with normal cell growth [4]. Cancer can also be described as a multiplex process that hinders the growth of normal cells, causes inexhaustible cell expansion, and changes the physiology of cells that can cause malignant cancer [5]. Studies of cancer have concluded that itis due to both environmental and genetic factors [6]. Physiological factors of cancer include hypoxia, insufficiency of genetic mutations, programmed cell death function, and oxidative stress, while apparent causes include pollution, radiation, and chronic exposure to ultraviolet rays [7]. The most common cancer among women is breast cancer which has a high death rate [8]. About 1.5 million cases of breast cancer occur worldwide each year [8]. About 1.7 million new instances were identified in the year 2012 [9].

The American Cancer Society estimated that about 281,550 new cases of breast cancer will be recognized in women in 2021. There are many kinds of breast cancer, but the most common are invasive carcinoma and ductal carcinoma in situ (DCIS). Others, such as angiosarcoma and phyllodes tumors, are less common [10]. The National Cancer Institute (NCI) has stated that avoidance is an effective means of reducing the number of individuals affected by cancer [11]. Most cancer preventive agents and anatomical administered breast cancer drugs are made to target only cancerous breast cells. Other healthy tissues may suffer from noxious effects. Studies concluded that flavonoids such as fucoidan could instigate cell cycle arrest [12]. Epidemic research and methodical examination have concluded that diets rich in vegetables and fruits are correlated with decreased risk of cancer originating in the epithelium such as mouth and breast. [13,14]. More intake of vegetables and fruits decreases the chances of cancer, including breast cancer [15,16]. Studies suggest that naturally produced compounds can have cancer preventive properties [17].

Flavonoids are polyphenolic compounds produced in plants in the form of bioactive secondary metabolites. Flavonoids consist of the flavone structure 15-C chain of phenylpropanoids. There are various types of flavonoids such flavanol and flavones. The capacity of flavonoids to restrain the metabolism of cells and to scrounge free radicals are reported in a number of studies [18]. Flavonoids are used in breast cancer treatment as antioxidants and by acting in macrophages by obstructing the formation of ROS (Reactive Oxygen Species), showing an immune modulating effect. The capacity of the flavonoids to select macrophages function to lower conditions such as immunosuppression of TMC in breast cancer [19].

Major research has been done on the efficacy of flavonoids such as quercetin, silibinin, apigenin, epigallocatechin-3-gallate, genistein, kaempferol, and naringenin against different cancer types, but due to poor absorption, low solubility, and easy metabolism, utilization of polyphenols in treatment of cancer is not sufficient. In this regard, current nanotechnology may be employed [20].

There are many types of polyphenols nanocarriers recently employed in the treatment of cancer. These include nanocapsules [21], solid lipid nanocarriers, nanoparticles [22], and metallic nanoparticles (gold) [23]. Selection of these techniques for nanoformulation leads to enhancement in biodistribution, pharmacokinetics, bioavailability, and specificity of drug transport to the site of cancer [24]. The applications of nanodelivery are discussed in Figure 1.

Among the naturally occurring compounds, flavonoids have been recorded to have anticancer properties [25,26,27]. Patients of breast cancer can be recognized by radiation therapy, hormone therapy, surgery, natural therapy, and chemotherapy. These remedies are used to control and prevent the development of metastatic tumors [28].

Studies suggest that the plant flavonoids show potential as antioxidants, with anti-inflammatory, anti-tumor, and atherosclerotic effects. There is interest in the chemoprotection action of dietary flavonoids in breast cancer, colon, rectum, and lung cancer. Flavonoids are used as a treatment in cancer by blocking the rapid increase of cancer cells by different mechanisms. A study of flavones and flavanols in post-menopausal women showed no links between flavonoid intake and breast cancer in pre- or post-menopausal women [29]. Breast cancer can be induced by DNA damage which occurs by metabolism, stress, and estrogen. A large number of flavonoids can improve breast cancer recovery by promoting the activity of cytochrome 1 enzyme in plants and animals [30].

Quercetin is a well-studied flavonoid for the treatment of breast cancer caused by estrogen and stress [31]. Recent reports of flavonoids as anticancer agents encourage the trials of flavonoids in humans. Phase 1 was performed on quercetin and results show that it suppressed the lymphocyte tyrosine kinase in 9 of 11 patients [32].

## 2. Pathophysiology of Cancer and Breast Cancer

Cancer is a diverse disease activated by the irreparable disabling of cells’ homeostasis and function. There are various signs that differentiate cancer cells from normal cells and lead to tumorigenesis such as proliferative signaling, metastasis, resting cell death, and angiogenesis [33,34] (Figure 2).

Three classes of genes are influenced in this process.

### 2.1. Oncogenes

Oncogenes are the genes which cause cancer. Proto oncogenes are converted to oncogenes which lead to cancer. RAS is an oncogene which acts as an on–off model in the transduction pathway. Mutations in RAS cause unrestricted tumor growth. This expression of unsuitable genes occurs at high levels in cancer.

### 2.2. Tumor Suppressor Genes

Tumor suppressor genes inhibit cell division by altering tumor development and any mutation in these genes causes hindering of normal cell growth and multiplication. In cancer patients these genes are damaged. In some studies, the first mutation is previously present in a cell of the germ line, thus all cells in the independent group inherit it because the mutation is introverted after that mutation arises in a second copy of the gene. Both the copies of genes are evolved, and cells exhibit uncontrolled growth. This results from changes in genetics [35].

### 2.3. DNA Repair Genes

A third category of gene related to cancer is a set of genes elaborated in the repairing of DNA and continuation of chromosome structure. Surrounding factors such as ultraviolet light (UV), chemicals, and ionizing radiation can harm DNA. Inaccuracy in DNA replication causes transformation [36].

### 2.4. ROS

Reactive oxygen species are free radicals, molecules, or ions with unpaired electrons in their outermost shell. Elevated levels of reactive oxygen species can result from increased metabolic activity, increased cellular signaling, activity of oxidases, paroxysm activity, and cross-links in infiltrating the immune system [37]. In addition, both radiation and carcinogens are possible sources of ROS that enhance single or double-strand breaks by acting with intra-strand cross-links, DNA sugar moiety, and protein-DNA cross-links [38].

### 2.5. Breast Cancer

Breast cancer is a malign tumor that initiates in the breast cells. There are diverse factors that can increase the probability of getting breast cancer.

Generally, two targets in the pathogenesis of breast cancer are:(1)Estrogen receptor (steroid hormone) that stimulates oncogenic pathways.(2)Epidermal growth factor [39].

Estrogen exposure can also cause breast cancer by damaging DNA and by showing mutation in the genes. Consequences of failure of immune defense can result in breast cancer. Sometimes it can also occur from defects in DNA or pro-cancerous genes such as BRC-A1 and BRC-A2.

## 3. Flavonoids Used in Breast Cancer

Flavonoids are categorized into various groups on the basis of carbon in the C ring, unsaturation, and oxidation of the C ring. The subcategories are discussed in Table 1:

### 3.1. Flavones

Flavones generally exist in leaves, fruits, and flowers such as glycosides [40]. The chemical structure of flavones shown in Figure 3. They show a number of activities such as antioxidant, antiviral, and anticancer properties. Flavones consist of apigenin, nobiletin, luteolin, tangeretin, chrysin, and baicalein. Flavones are widely studied as anticancer agents and work by inhibiting tumor enlargement in the cells of cancer by the process of apoptosis [42]. They have no adverse effects.

In human cancer cells including prostate and melanoma, luteolin exhibits arrest of the cell cycle whereas dis-arrest produced by luteolin is associated with suppression of CDA2 activity in colorectal cancer. Luteolin also represses the function of the epidermal growth factor (EHFR) breast cancer cell line in humans [43]. Luteolin decreases the progression of MPA-dependent breast cancer cell allograft.

Luteolin initiates the process of apoptosis in breast cancer cells by the inhibition of fatty acid synthesis which is an essential lipogenic enzyme overexposed in many human cancers [44].

### 3.2. Flavonols

Flavonols are the main sub-category of flavonoid, that have the 3-hydroxyflavone backbone shown in Figure 4. They are found in onions, tea, some common fruits, and broccoli. Flavanols such as isorhamnetin, kaempferol, quercetin, and myricetin are some examples [45]. Flavonols may inhibit breast carcinogenesis. Flavonols may be pro-oxidant, antioxidant, anti-estrogenic, or modulate of cell signaling pathways. A recent analysis of flavonoid consumption and breast cancer recommended that flavonols and dietary flavonoids are linked with a lower risk of breast cancer, mainly in post-menopausal women [46].

Quercetin is a glycoside found mainly in onions, apples, and garlic can initiate damage of DNA in cancer cells [47]. Current research shows that quercetin plays a pivotal role in inhibiting transduction of signals, apoptosis in cancer cells and by suppressing the multiplication of tumor cells [48,49,50]. Kaempferol plays an important role in the inhibition of growth of breast cancer cell lines. Flavones are a major class of flavonoid and are mainly found in citrus fruits such as lemons, grapefruit, and oranges. Naringenin, eriodictyol, and hesperidin are examples of flavanones. Flavanone containing citrus has pharmacological effects such as anti-inflammatory, cholesterol lowering, antioxidant, and blood lipid lowering activity [51]. Citrus juices are procured from blond oranges, grapefruit, mandarins, sour oranges, lemons, tangors, bergamots, tangerines, limes, kumquats, etc., the major constituent in blond oranges is 200–600 mg/L of hesperidin but they also contain didymin (19–35 mg/L) and narirutin (16–84 mg/L). Hesperidin is also the predominant compound in lemon juice and lime [52].

Hesperidin, an inhibitor of aromatase and a main curative agent for estrogen receptor positive breast cancer in perimenopausal women has also shown to be an inhibitor of enlargement of MDA-MB231 cells, probably due to glucose uptake deterioration and repression of glucose transporter1(GLUT-1). Narigenin, also a flavanone, is a weak estrogen and appears to be used in lowering the P38 MAPK pathway. It suppresses metastasis of lung in breast cancer patients after surgery by stimulating immunity in the host by an extension of Interleukin (IL-2) T cells and Interferon (INF)-alpha [53].

### 3.3. Catechins

Catechin is a flavonoid which was mainly found in tea leaves. It comprises a set of polyphenols that contains catechin various health benefits. Epigallocatechin gallate (EGCG), epicatechin (EC), epigallocatechin (EGC), and epicatechin gallate (ECG) are the vital catechins found in tea leaves [54]. In vitro and epidemiological studies of catechin associated with green tea produce depression in breast cancer patients [55]. Consumption of green tea was not related to the general risk of breast cancer [56]. However, the reaction of tea and its catechins on the diagnosis of breast cancer are still unknown [57,58].

Tea and tea polyphenol have been studied for the last two decades with emphasis on the prospective cancer chemo preventive and medicinal properties of teas. Present data suggest a strong chemopreventive and therapeutic effect of green tea polyphenols and EGCG against cancer of the skin, breast, liver, prostate, colon, lung, and liver [59]. In Asian countries, epidemiological scrutiny has recommended that the lower occurrence of some cancers is because of the consumption of green tea [60].

### 3.4. Isoflavones

Isoflavones are polyphenolic and they mainly present in soybeans which area rich source of protein for millions of individuals. The chemical structure of isoflavones shown in Figure 5. Soybean contains isoflavones, which are phytoestrogen polyphenols with vigorous anticancer effects. Soybean rich isoflavones are an anticancer compound, soy foods contain many bioactive compounds such as lunasin, omega-3-fatty acid, Bowman–Birk inhibitor, and dietary fiber. Soy foods are used in the prevention of diabetes, cardiovascular disease, hypertension, and cancer [61,62,63,64].

Daidzein, genistein, and biochanin A are formononetin phytoestrogens. A rich source of isoflavones are legumes from the family Fabaceae [65]. Soybean (*Glycine max*), such as glycitein, daidzein, genistein, and red clover (*Trifolium pratense*), has an etymology from biochanin A and formononetin. Daidzein and genistein are the most known isoflavones because of their estrogenic effect in animal models.

### 3.5. Anthocyanidins

Anthocyanins are abundantly found in plants and are a category of flavonoids. They are present in various fruits and flowers such as grapes, strawberries, and corn, and can exhibit purple, blue, or red color. Anthocyanins have a number of properties such as bacteriostatic, anti-aging, anticancer, and anti-inflammatory activities.

They work by inducing differentiation, a process in which malignant cells differentiate into normal mature cells by the inducers [66].

Anthocyanins can stop the proliferation of cancer cells in three ways:(1)Signal pathway inhibition for blocking of signal transduction.(2)Regulation of antioncogenes.(3)By acting on beta-catechin, Notch pathway.

Anthocyanidin and its extract have manifested anti-cell multiplication outcomes regarding distinct types of breast cancer cells [17,18]. Examples include, Anthocyanins isolated from Mix Eco Blue corn, which retards the proliferation of cells by arresting the cell cycle at G1 phase.

Another study shows an extract of anthocyanins from strawberry inhibited the multiplication of cell on breast cancer cell lines in murine [18].

They were initiated to inhibit cancer cells by lowering the expression of Matrix Metalloproteinase (MMP) and urokinase plasminogen activator, which deteriorate the extracellular matrix as an invasion process by restoring the expression of inhibition, both of which nullify MPA and MPP action [67].

In one study, when treatment was done with P3G and C3G sensitive Transtuzumab resistant MDA-M-53 R which initiates process of apoptosis, inhibition of p-HER 2 and P Marks took place and showed inhibition of human breast cancer in vitro [68].

### 3.6. Procyanidins

Procyanidin is a class of flavonoid belonging to proanthocyanidin or condensed tannins. Procyanidin exists in trimers, dimers, and tetramers and any oligomers of epicatechin and catechin shown in chemical structure (Figure 6). Dimeric procyanidins exist as procyanidin A1, B1, A2, B2, B5, B3, B6, B8, and B5; and trimericsC1, C2; procyanidins, and tetrameric procyanidin are cinnamtannin A2 and catannin A2. Procyanidin mostly present in cocoa, apple, berries, and grapes at elevated concentrations [69].

The prime source of proanthocyanidins are fruits and berries. Cranberry, black chokeberry, black currant, lingonberry, and black elderberry are some consumable berries contain proanthocyanins [70]. Raw banana, Chinese quince, persimmon, and carob bean have an astringent effect because of the presence of proanthocyanidin. As per research, high amounts in fresh individual fruit were discovered in rose hips, cocoa products, and chokeberries [71,72]. Grape seed proanthocyanidin (GSPs) are taken as dietary nutrients.

Proanthocyanidin rich grape seeds are broadly ingested as a dietary nutrient and have preventive action against cancer as well as chemotherapeutic activity in different culturing of cell and animal models [73]. Recent studies reveal that GSPs have restrictive effects in vivo and in vitro on different cancers without harmful results on normal cells [74,75]. However, it is not clear if GSPs have efficacious chemopreventive effects against breast cancer.

### 3.7. Lignans

Lignans are plant components that are absorbed in the gut to form enterolactone, phytoestrogens and enterodiol. The configuration of the lignans was unknown till recently: preliminary studies have been performed based on the possibility of urinary excretion of lignans metabolites reduced the risk of breast cancer related to higher excretion of lignan enterolactone and estradiol [76].

Lignans possess anticarcinogenic effects and exposure high levels may be linked with reduced chances of breast cancer [77]. Lignans show dual function as antioxidants and as phytoestrogens. Flaxseed has nearly 800 times more lignan than any other food. Lignan exhibits functions in the curing of diseases including cancer. They consist of SDG, lariciresinol, matairesinol, and pinosterol.

In one of the studies, cell growth reagent WST-1 was used on AML cell lines, Monomac and KGI analyzed with END, SDG, and ENL to evaluate the cytotoxic effect of flaxseed lignans on cells. The outcomes appear to show that END and SDG have a minimum antiproliferative effect on KG-1 cells but ENL revealed a dose dependent sequel on Monomac-1and KG-1 cells [78].

Flaxseed lignans also exhibit antitumor activity for hormone sensitive cancers. One study shows that the dietary consumption of flaxseed in the case of mice provided with estrogen receptor (ER) negative human breast cancer, lowered the growth of tumor and metastasis in the downregulation process of insulin-like growth factor [79].

### 3.8. Flavan-3-ol

Flavan-3-ol are a vital group of nutritional bioactives which belong to the category of polyphenolics present in pome fruits, cocoa-derived products, tea, berries, and nuts. Studies show that consumption of dietary food rich in flavan-3-ol enhances improves vascular function in healthy adults [80].

In the United Kingdom, Australia, and Japan with the culture of tea, tea is the major constituent of flavan-3-ol [81], although in other countries are fruits, specifically pome fruits [82]. This variance leads to much alteration of the variety of flavan-3-ol (Figure 7) consumed.

Flavan-3-ol comprised of (+)catechin, (−)epicatechin, (−)epigallocatechin 3-gallate, theaflavin 3,3′-digallate, (+)gallocatechin, (−)epicatechin 3-gallate, theaflavin 3-gallate thearubigin, theaflavin 3-gallate, and theaflavin 3′-gallate [83], which are present in wine, cocoa, tea, and apples. A large study showed that flavan-3-ol expresses a biological action both in vivo and in vitro by functioning as an anticarcinogenic, anti-viral, antioxidant agent, neuroprotective, anti-microbial and cardio-preventive agent [84].

Green tea, a beneficial source of catechin, is used to make more absorbable structures to cure different kinds of cancer, including breast cancer. A complete examination for EGCG was made in various studies. As per Xenograft studies, EGCG with tamoxifen was likely in ER-negative breast cancer models. The MDA-MB-231-mediated tumor capacity was reduced while administering 25mg/kg of EGCG/EGCG + tamoxifen in an athymic naked mice model [85].

## 4. Role of Flavonoid in Breast Cancer Control

Cancer is a multistage disorder which includes environmental, physical, genetic, chemical, and metabolic factors. A diet high in vegetables and fruits which are rich in antioxidants decreases the chances of many types of cancers, with antioxidants suggested to be efficacious [86]. Chances of developing breast cancer affected by non-adaptable factors which include genotype and age, and adaptable factors such as alcohol consumption, occupational exposure, smoking, and nutrition [87]. Obesity, inactive lifestyle, and harmful foods are the factors that increase mortality rate of breast cancer patients. Whether consumption of polyphenols from diet prevents breast cancer is debatable, as only high amounts of polyphenols are adaptable to hinder ERalpha+BC and ERalpha cells, while decrease in levels of polyphenols can restore ERalpha+ cells growth [88].

Breast cancer mostly consists of ER negative and ER positive cells; accordingly, agents which contain a preventive action on both ER-negative and ER-positive cells can be effective against tumors [89]. Naringenin, a flavonoid which is present in citrus fruits, produces a relatable cytostatic effect in human BC cells. High consumption of flavones, available in aromatic plants, has been correlated with lower the chances of BC [90].

Epidemic research executed on human populations mostly focuses on daidzein and genistein, suggesting that isoflavones present in soy foods are preventative against BC [91,92]. Kaempferol acts as a cancer preventive agent. It was developed to suppress the growth of different tumor cells such as leukemia (HL-60 and Jurkat), glioblastoma (LN229, U87MG and T98G), prostate cancer (LNCaP, PC-3 and DU145), lung cancer (H460 and A549), pancreatic cancer (M1A PaCa-2, PanC1), and specifically breast cancer (MCF-7, BT-549 and MDA-MB-231). It apprehends the cell cycle in tumor cells. Kaempferol is mostly efficacious against angiogenesis [93]. The role of flavonoids in the treatment of breast cancer is explained in Figure 8.

Quercetin is a dietary flavonoid that inhibits the growth of tumors by hindering the protein tyrosine kinase (PTK). Diosmin exhibits antitumor activity in identical cancer cell lines. Tangeretin is a flavone that produces superior antitumor activity to other polyphenols against SK-MEL-5, SK-MEL-1, and B16F10 melanoma cell lines [94,95], PLC/PRF/5 cell lines, human lung DMS-14 cell lines, human hepatoma HepG2, Hep3B, and in breast MDA-MB-435 and MCF-7 cell lines. Flavone, including luteolin, appears to hinder cell growth and cell proliferation of various types of cancer which includes prostate cancer [96], hepatocellular carcinoma, pancreatic cancer cells, and non-small cell lung cancer cells. Flavonoids such as flavones and flavonols show antagonistic effects with luteolin against cervical cancer, human colorectal adenocarcinoma, prostate cancer cell, and breast cancer cells [97].

An anthocyanin-rich extract from black rice (AEBR) has a role against breast cancer [98]. Flavanones act as inhibitors of distinct types of cancer cells which influence apoptosis. Both in vivo and in vitro studies concluded that flavanones such as naringin, 2′-hydroxyflavanone(2-HF), and hesperidin inhibit the growth of cancer cells and promote programmed cell death of cancer cells, the involvement of mitochondrial and caspase-dependent pathway, and death receptors. 2HF has been tested for its anticancer activity in human lung cancer cells, colon, and breast cancer cells [99,100].

## 5. Nano-Based Formulation of Phyto-Bioactive Used in Breast Cancer Prevention

Currently, there has been extensive growth in the area of nano based delivery to transport innate forms of vital substances and medicinal agents for the therapy of different diseases [101].

One main complication in the study is that expanding multipurpose nanomolecules have effects which permit them to carry individual curatives across different biological hurdles and are suitable to mark specific cells, organs, and tissues in the body. Between different nanodelivery systems, there are both polymeric nanocrystals, solid types such as lipids and nanoparticles, and liquid types, such as nanoemulsions, nanopolymers, and nanoliposomes [102]. There are different nanomaterials that are used in the delivery of drugs. These nano phyto-bioactive and their properties are described below.

### 5.1. Nano-Curcumin

Curcumin is a diarylheptanoid, polyphenol (shown in Figure 9) extracted from the root of the turmeric plant, acquired from ferulic acid, and segregated from the Indian plant *Curcuma longa* (Zingiberaceae). It has antineoplastic, antifungal, and antibacterial properties. US FDA(GRN-686) have approved this plant as Generally Recognized as Safe (GRAS) [103].

The use of curcumin in the targeting and prevention of different aging related pathological situations has been frequently recorded [104]. These states consist of inflammation, cardiovascular disease, oxidative stress, type-2 diabetes, rheumatoid arthritis, ocular disease, atherosclerosis, osteoporosis, neurodegenerative diseases, and cancer. From the last decade, the possibility of this compound to enhance therapeutic effectiveness has been examined in clinical trials [105].

Along with therapeutic activities, a vital problem with curcumin is the bioavailability due to its low solubility in aqueous solution which limits clinical efficacy. To overcome the problem, nanocarriers for curcumin have been developed. Overdose of drugs is still a common clinical issue across the world, taking place both intentionally and unintentionally. Antidotes exist, but are not easily available. Therefore, nanoparticles have been formulated for delivery of many drugs [106].

Clinical trials have been performed for the treatment of different types of cancers such as pancreatic, prostate, and breast cancer. The various nano flavonoids used in the treatment of breast cancer are described in Figure 10.

### 5.2. Nano-Epigallocatechin-3-Gallate

Epigallocatechin-3-gallate [EGCG], a polyphenol(catechin) obtained from green tea, has a role as an anticancer drug [107]. It also acts as an anti-HSP90, antioxidant, neuroprotective, and neoplastic agent [108]. This medication is being clinically tested in different types of cancer therapies including lung cancer (Phase 2), prostate cancer (Phase 1), and colorectal cancer (Phase1) (NCTO2577393, NCTO459407, NCTO2891538).

An epidemiology investigation was performed to scrutinize repeatedly contrasting in vitro detecting. This uncertainty could be due to its poor bioavailability and stability. Hence, to enhance bioavailability of EGCG, new nanodelivery systems have been employed. Elevated potential and more stability for oral conveyance were available in EGCG-loaded solid lipid nanomolecules than for untreated EGCG [109].

Nanoparticles that enhance the anticancer properties of EGCG are chitin-loaded honokiol and nanoethosomes. An investigation proved that delivery of docetaxel present in EGCG-nanoethosomes transdermally diminish the volume of tumor. The nanoethosome permits flavonoids’ transdermal delivery to melanoma cancer cells [110].

### 5.3. Nano-Naringenin

Naringenin (NAR), a naturally present flavonoid in plants such as citrus fruits, has a great number of pharmacological activities. Its scientific occurrence has been hampered because of poor aqueous solubility and improper transport across biological membranes which lead to low bioavailability at the site of the tumor [111]. Its chemical structure shown in Figure 11.

In contemporary studies, antiproliferative outcomes of Naringenin with cyclophosphamide were studied in breast cancer cells and involvement of JAK-2 and STAT-3 were explored. It is seen that naringenin also has the ability to repress the functions of IL-6 in regulating apoptosis-associated expression of genes. Naringenin and cyclophosphamide can impair the uncontrolled division of cells to a sustainable degree and can be distributed as potent chemotherapeutics for the treatment of breast cancer [112].

Many nanoscaled delivery systems are used, such as polymeric nanoparticles, nano suspension, and nano emulsions. Nano emulsions have an elevated ability to alter the bioavailability of drugs which are less soluble by oral route. To increase the oral bioavailability and solubility of naringenin, some papers describe the use of nanoemulsions as a system of drug delivery for NRG [113].

Nanonargenin works by inhibiting both PI3K and MAPK paths and by restricting ER alpha to the cytoplasm to lessen proliferation of Tam-RMC cells. This study demonstrates the mode required in the proliferation of cells of Tam-R breast cancer cells [114].

### 5.4. Nano-Quercetin

Quercetin (QT) is a pentahydroxyflavone polyphenols shown in Figure 12. It is found in red grapes, olive oil, bracken fern, lettuce, onion, tea, coffee, and citrus. QT has noxious effects on various cancer cell types. However, lesser bioavailability and deficient solubility of QT have restricted its therapeutic use [115]. QT has cytoprotective effects to hinder various molecules that are used in some signaling pathways, imparting carcinogenesis activity, and containing protein kinase-C, tyrosine kinase, and phosphatidylinositol-3 kinase. Modern research exhibits the importance of tyrosine kinase pathways in breast cancer continuance [116]. The antioxidant polyphenol quercetin can limit damage from free radical. However, quercetin, an inhibitor of lipid peroxidase, might be helpful as an anticancer cell [117]. Hence, it is demonstrated that quercetin acts as an inhibitor in the growth of the MCF-7 cell line.

Metabolism and maintenance of blood levels in tissues is accomplished by encapsulation of quercetin for a prolonged period of time. It enhances bioavailability, solubility, and increased sustained release by nanoencapsulation, and may improve the bioactive quercetin-loaded nanomicelles that are stable in intestinal and gastric fluids [118].

### 5.5. Nano-Resveratrol

Resveratrol is a 5,4′-Trihydroxystilbene (Figure 13), natural polyphenol, has a range of biological properties, including antioxidant, anti-fungal, anti-aging, anti-inflammatory, and antiviral activities. Several studies have proven that RES contains chemoprotective activities such as neuroprotective- and cardioprotective effects [119]. Administration of resveratrol with chemotherapeutic agents reduces the toxic effects associated with it and enhances the medicinal efficacy related with chemotherapy of cancer.

Beta glucan has been utilized as a complementary carrier of drugs, or in addition with a drug such as resveratrol by using a drug delivery system to increase bioavailability. Research has recognized the synergistic effects of resveratrol and beta glucan on immune reactions. Combinational effects were examined in each case, and the potential of these compositions on gene countenance (such as Cdc42, NF-KB2 and BcL-2) in cancer cells of breast were analyzed [120].

### 5.6. Nano-Apigenin

Apigenin is a 4′,5,7-trihydroxyflavone as shown in Figure 14. It is a flavonoid having low molecular weight, found in plant-derived beverages, fruits, nuts, and vegetables that hinders the growth of cancer cells of humans in vivo and in vitro [121]. Apigenin has shown effects on different types of cancers such as liver cancer, prostate cancer, colorectal cancer, lung cancer, melanoma, osteosarcoma, and breast cancer [122].

This polyphenol is used in modulating signaling pathways in skin cancer and hepatocellular carcinoma. The favorable effect of individual apigenin in the treatment of cancer is comparatively low due to its low water solubility [123]. These days, to enhance the polyphenols bioavailability, PLGA nanoparticles are mostly used.

Research shows that the bioavailability of apigenin is low. To enhance the functionality of apigenin, a nanoformulation of apigenin is under examination in in vivo and invitro models. Biosensors and gold nanoparticles (AuNPs) used in photothermal therapy have acquired marketability because of their attainable implementation in drug delivery and in the treatment of cancer [124].

Apigenin-AuNPs also show adequate antiangiogenic properties. Moreover, studies on nano-apigenin utilizing poly(lactic-co-glycoside) (PLGA) are efficacious in topical and oral administration for skin care, attaining a higher potency and efficacy with decreased toxicity. This evidence proves that nanoformulations of apigenin could enhance an essential drug-delivery system.

### 5.7. Nano-Silibrin

Silibrin is a flavonolignan obtained from the fruit of *Silybum marianum* Gaertner. Silibrin-loaded nanoparticles suppress tumor angiogenesis and the epithelial–mesenchymal transition (7). Silibrin and cryptotanshinone effectively perforate the barriers of the intestine, hence increasing the pharmacokinetic of drug loads [125].

These nanoparticles manifest beneficial anti metastasis resulting in breast cancer bearing nude mice. In order to get better results against metastasis of breast cancer, lipid nanoparticles loaded with silibrin consisting of d-alpha tocopheryl polyethylene glycol 1000 succinate, silibinin, and phosphatidylcholine were obtained [126].

Silibrin loaded nanoparticles (SLNs) can accumulate in tissues of tumor in a planned order. In comparison to free silibrin, SLNs contain many inhibitory effects in the conquering of MDA-MB—231 cells by down regulation of snail and MMP.

Nano Isomers filled with silibinin can be used as feedback in breast cancer. In one of the studies, silibinin was encapsulated in glycol and its effectiveness against cancer was checked [127].

### 5.8. Nano-Genistein

Genistein is a natural isoflavonoid of phytoestrogen class, abundantly found in soy, which has breast-cancer preventive properties. A recent study was conducted revealing the action responsible for the inhibitory activity of genistein in MCF-7 cells. It arrests the growth of MCF-4 cells at the M/G2 phase and lowers it at the proliferative S-phase [128].

Genistein-loaded PEGylated silica hybrid nanomaterials were produced by an efficient and simple aqueous dispersion method. Infrared analysis also found that the encapsulation of genistein was fully obtained with effectiveness in breast cancer treatment [129,130]. Various nanoformulations of flavonoids in the treatment of breast cancer with MOA are detailed in Table 2.

## 6. Conclusions

Evidence suggests that a diet rich in polyphenols present in vegetables and fruits can decreases the risk of different cancers. Moreover, phytochemicals found in plants work as supportive medicine to suppress and preserve the growth of different types of human cancers. Several studies have been shown to have a positive effect in modulating immune response, supporting and replacing the common functions of cells, and reducing swelling. Flavonoids can be used in the treatment of breast cancer. Flavan-3-ol, procyanidin, isoflavone, flavone, flavanol, flavanone, lignans, anthocyanidin, and catechin all have anticarcinogenic properties and are used in the prevention and treatment of breast cancer. Their activity depends upon type of cell used. Natural polyphenols have different anticancer properties, including cancer prevention agent enzymatic movement, inhibition of cell proliferation, and development of apoptosis. The effects of polyphenols as anticancer drugs differ with the type of cancer, doses, and cell lines. In addition, many studies should be performed with new methods such as the modification of chemical structure of flavonoids that would enhance effectiveness in shielding of damage to DNA and bioavailability to increase efficacy of flavonoids.

The current review briefly discusses nano-delivery of phyto-bioactive compounds. Extensive study is being performed on nano drug delivery systems to increase the bioavailability of herbal drugs and protect from GI deterioration. However, herbal therapy requires a scientific approach to transport the constituents in a novel manner to enhance patient compliance and increase oral administration of phyto-bioactive compounds. Therefore, nanotechnology gives an ideal carrier system to enhance the bioavailability and pharmacokinetic profile of flavonoids.

## Figures and Tables

**Figure 1 molecules-26-05163-f001:**
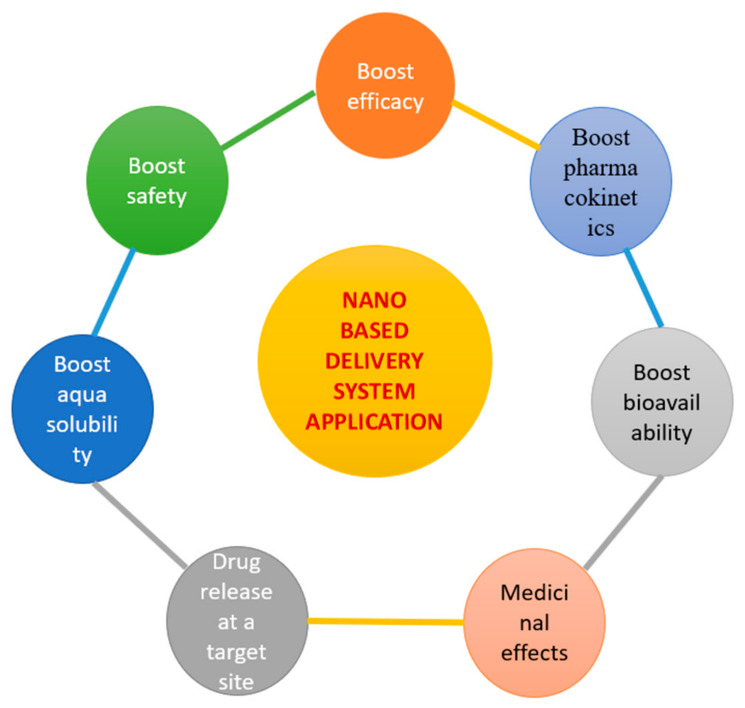
Benefits of nanodelivery systems.

**Figure 2 molecules-26-05163-f002:**
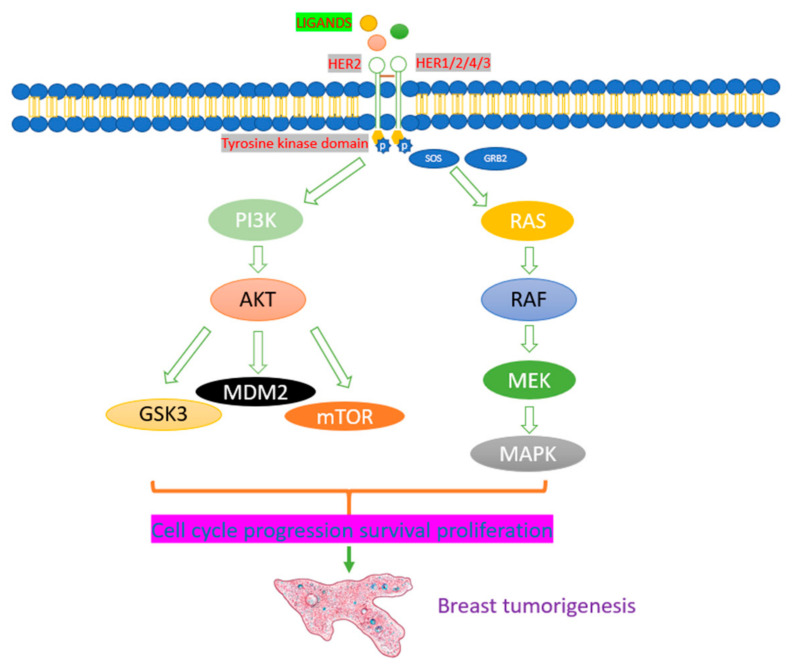
Pathophysiology of breast cancer. HER2 and other tyrosine kinase receptors present on the cell membrane and join to ligands. Tyrosine kinases undergo phosphorylation and start signaling pathways such as AKT and RAF.

**Figure 3 molecules-26-05163-f003:**
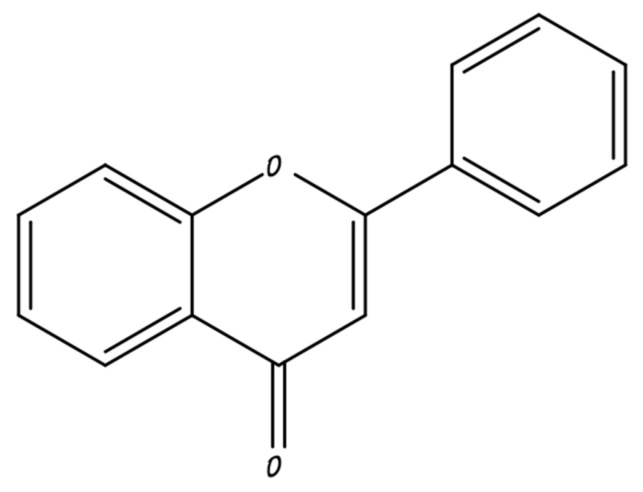
Flavone.

**Figure 4 molecules-26-05163-f004:**
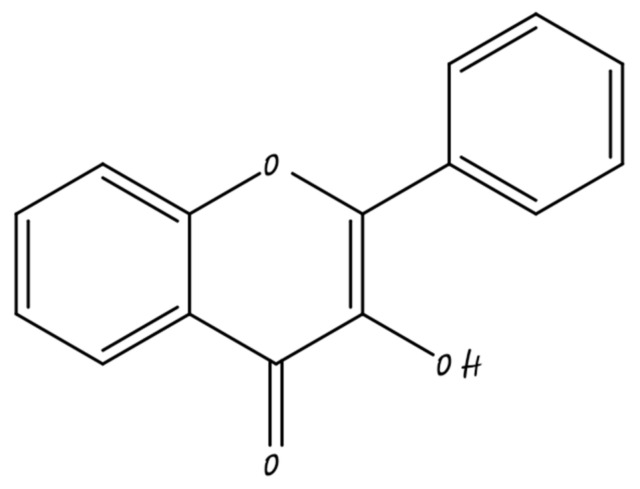
Flavonol.

**Figure 5 molecules-26-05163-f005:**
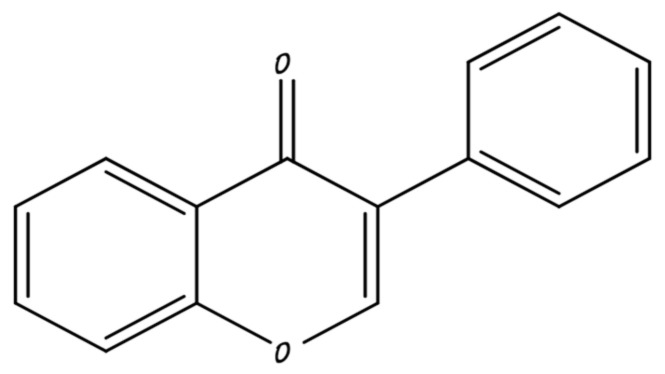
Isoflavone.

**Figure 6 molecules-26-05163-f006:**
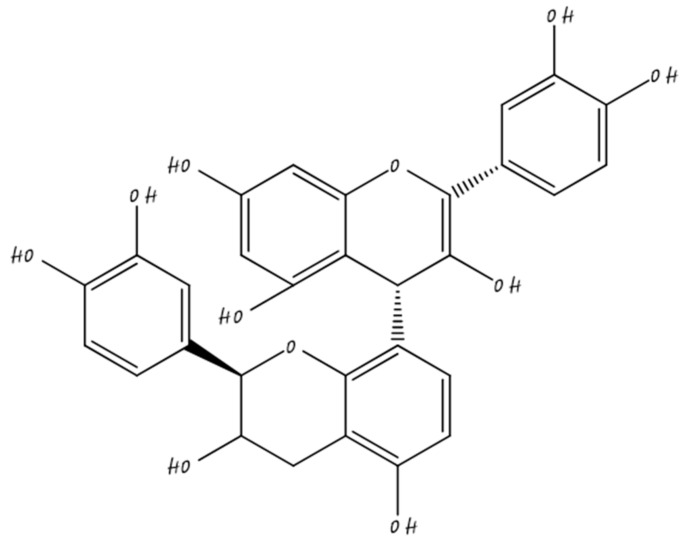
Procyanidin.

**Figure 7 molecules-26-05163-f007:**
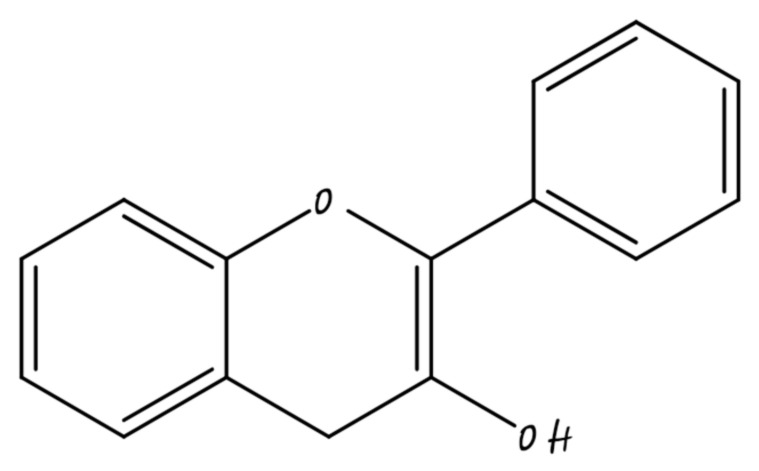
Flavan-3-ol.

**Figure 8 molecules-26-05163-f008:**
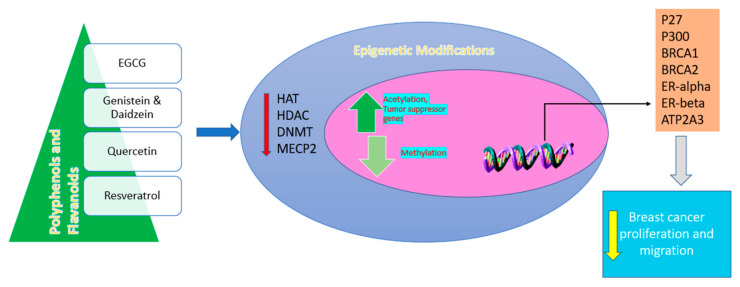
Diagram depicts epigenetic modification and tumor suppressor genes’ induction by the mechanism of flavonoids.

**Figure 9 molecules-26-05163-f009:**
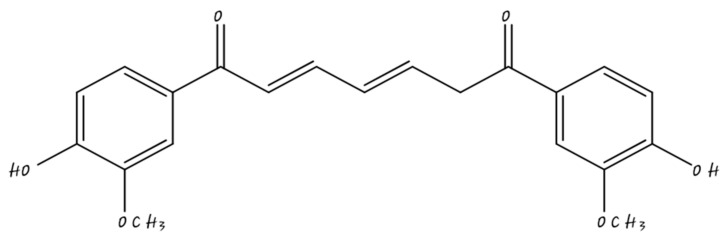
Curcumin.

**Figure 10 molecules-26-05163-f010:**
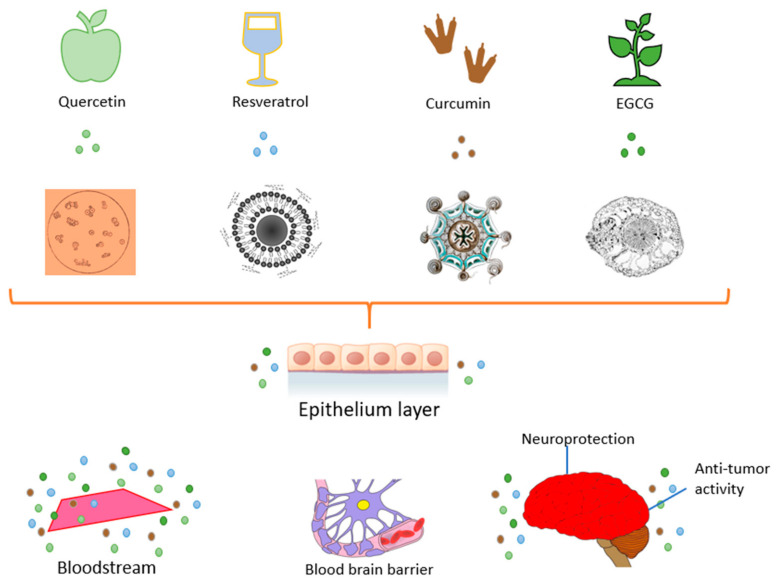
Therapeutic potential of nano flavonoids on breast cancer.

**Figure 11 molecules-26-05163-f011:**
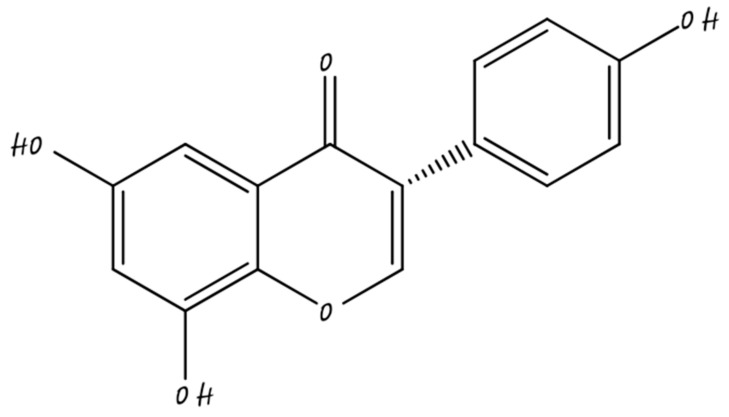
Naringenin.

**Figure 12 molecules-26-05163-f012:**
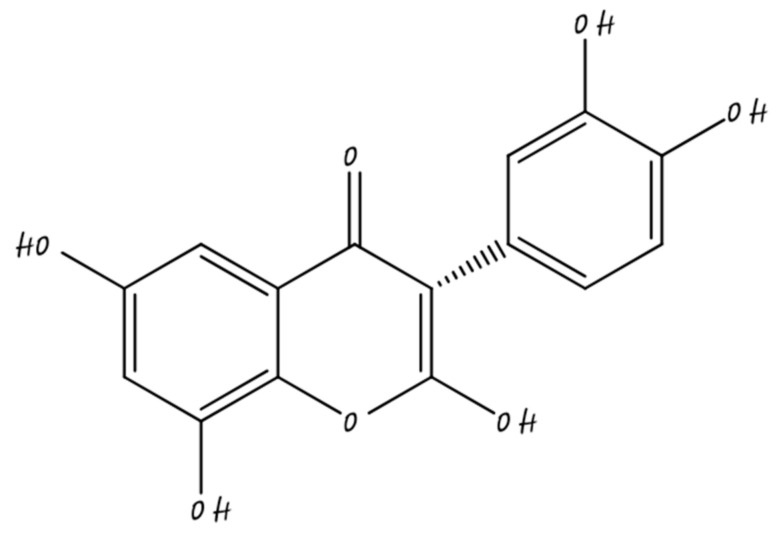
Quercetin.

**Figure 13 molecules-26-05163-f013:**
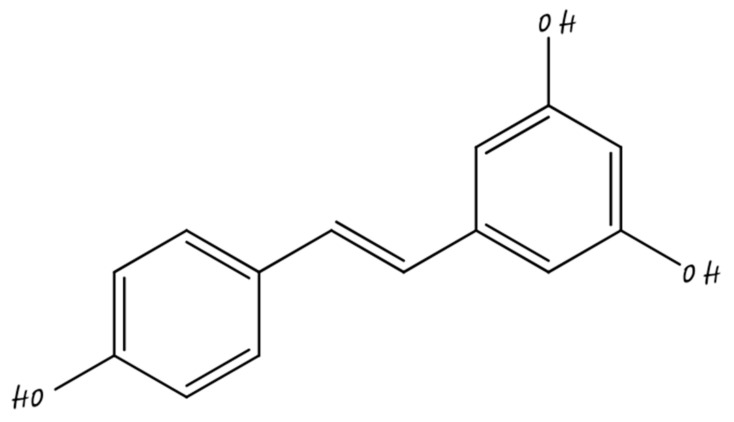
Resveratrol.

**Figure 14 molecules-26-05163-f014:**
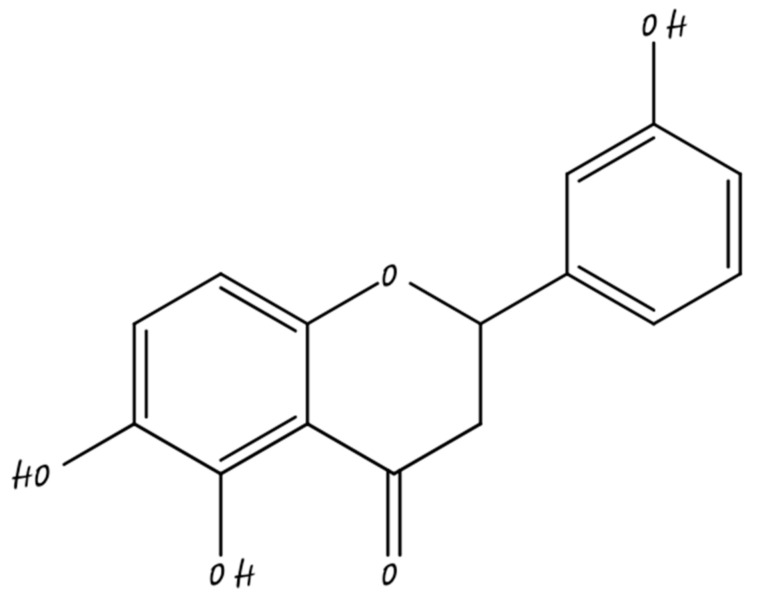
Apigenin.

**Table 1 molecules-26-05163-t001:** Various types of flavonoids.

NO.	FLAVONOIDS	CONSTITUENTS
1	FLAVONES	Consist of apigenin, tangeretin, luteolin, chrysin [39]
2	FLAVONOLS	Consist of quercetin, isorhamnetin, myricetin, kaempferol [40]
3	FLAVONOIDS	Consist of naringenin, eriodictyol, hesperidin
4	CATECHINS	Consist of epicatechin, epigallocatechin gallate, (−)-epicatechingallate (ECG), (−)-epigallocatechin found in tea leaves
5	ISOFLAVONES	Consist of biochanin A, phytoestrogens, genistein, formononetin, daidzein
6	ANTHOCYANIDINS	Pelargonidin, cyanidin, malvidin, delphinidin, peonidin, petunidin
7	PROCYANIDINS	These are the oligomers of epicatechin, gallic acid, esters and catechin
8	LIGNANS	Plant components absorbed in gut to form phytoestrogens, enterolactone, enterodiol
9	FLAVAN-3-OLS	Consist of catechin, epicatechin, epigallocatechin, theaflavin-3-gallate, theaflavin-3,3′-digallic acid [39,41]

**Table 2 molecules-26-05163-t002:** Nanoformulations of flavonoids with mechanism of action in treatment breast cancer.

NO.	NANOFORMULATION	MECHANISM OF ACTION
1	NANO-CURCUMIN	Use of curcumin in targeting and prevention of different aging related pathological situations has been recorded [103]
2	NANO-EPIGALLOCATECHIN-3-GALLATE	Delivery of docetaxel present in EGCG-nanoethosomes transdermal diminish the volume of tumor [110]
3	NANO-NARINGENIN	Works by inhibiting both P13K and MAPK paths also restricts ER alpha to lessen the proliferation of tam-RMC cells [114]
4	NANO-QUERCETIN	It acts as growth inhibitor of MCF-7 cell lines
5	NANO-RESVERATROL	Combinational effect of resveratrol and Beta-glucan potentiates the composition on gene countenance (such as cdc42, NF-KB2 and BCL-2) in breast cancer cells [120]
6	NANO-APIGENIN	Biosensors used in photothermal therapy and have acquired implementation in treatment of cancer [121]
7	NANO-SILIBRIN	Silibrin loaded nanoparticles (SLNs) contain inhibitory effects on conquering of MDA-MB-231 cells
8	NANO-GENISTEIN	Responsible for the arrest of growth of MCF-4 cells at M/G2 phase and lower at proliferative S-phase [125]

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
