# Peer review of "Impacting the Remedial Potential of Nano Delivery-Based Flavonoids for Breast Cancer Treatment"

_molecules, 2021, doi:10.3390/molecules26175163_

Round 1
Reviewer 1 Report
The manuscripts is devoted to nano delivery-based flavonoids for breast cancer treatment. In general, you did not get deep into the important concepts that need to be highlighted in this review. Please, the following issues need to be amended before publication.
1. The title did not represent the review content. There is only a short sentence about nanodelivery in the abstract, and much more about the function and benefits of flavonoids in breast cancer treatment.
2. Antiestrogen are also provided to women who are recognized with breast cancer, especially estrogen receptor positive tumors. Breast cancer appears to show intratumoral heterogeneity with estrogen receptor positive and negative cells.
This part seems unfinished since it is not clear why the authors are bringing in this context.
3. It's better to introduce breast cancer first and then to introduce flavonoids used for breast cancer treatment
4. These includes nanocapsules [4], solid lipid nanocarriers, nanoparticles [5] and metallic nanoparticles (gold) [6].
Multiple characteristics of each kind of NP drug delivery system need to be elaborated in this content.
5. Figure1 appear cluttered. There is some redundancy like drug release at a target site boost ability of target. And I think this figure shows the benefits of nano-delivery system instead of its application.
6. Cancer is an inherited disease which is a crucial health complication over the world [7].
What do the authors imply by cancer is an inherited disease? Please elaborate.
7. Figure 2 and 8 need figure legend, readers can not understand the data presented without figure legend
8. Lines 127-142
Can you elaborate better on how this three classes of genes and ROS function in cancer? Without those explanations, these lines make less sense because of the lack of rationale.
9. The tables contains too little information.
10. 3. Flavonoids used in Breast Cancer 3. Flavones
There reappear 3.
11. They have no adverse effects.
Is there any study to confirm that this drug has no side effects at all? Please cite appropriately.
12. Figure3-7 could be put together into one figure.
13. Multiple citations missing, such as line 48-50, 53, 54, 56, 128, 129, 187…
Author Response
|
S.NO |
COMMENTS |
REPLIES |
|
1. |
There are only short sentences about nano delivery in the abstract. |
As per the comment, nano delivery, along with examples, has been elaborated in the abstract. |
|
2. |
Its better to introduce breast cancer first and then introduce flavonoids used for breast cancer treatment. |
As per comment, breast cancer has been introduced first and then flavonoids in their treatment. |
|
3. |
Nanoparticles like nano capsules, solid lipid nanocarriers and other metallic particles need to be elaborated. |
As per comments, Nanoparticles are elaborated. |
|
4. |
Figure 1 appears cluttered.
Figure 2 and 8 need figure legends. |
As per comments, changes have been done in the figures. Figure legends has been added. |
|
5. |
How cancer is crucial health complication over the world |
As per comments lines has been elaborated. |
|
6. |
Elaborate better on how classes of gene and ROS function in cancer |
As per comments, Classes of genes and ROS function has been elaborated. |
|
7. |
Is there any adverse effects of drugs in the manuscript |
Yet not particular Adverse effects has been confirmed (breast cancer) . |
Reviewer 2 Report
This review describes the numerous factors which contribute to the insurgence of breast cancers and propose the beneficial properties of flavonoids as a possible remedy. The subject is relevant and up to date, and the description of the different classes of polyphenolic compounds is very accurate. In addition, the schemes and tables presented by the authors are useful in helping the reader to follow the various topics illustrated. However, the level of english language is very poor and many non appropriate terms are used thoughout the manuscript. Moreover some parts of it are ill-structured and present repetitions, abrupt changes of subject, or incomplete development
In my opinion the entire structure of the review should be revised. I made some sugestions for possible revisions in the astract and introduction, just to show what I mean. Curiously enough, the topic nanodelivery is not resumed and commented in the conclusions, while it appears to be a relevant aspect of the review. This is also the topic for which the referencies given should be increased, since several important stuies have not been cited.

Author Response
|
S.NO |
COMMENTS |
REPLIES |
|
1. |
Entire structure of the review should be revised |
As per comments, structure of review has been revised. |
|
2. |
Conclusion |
As per comments, nano delivery has been elaborated in the conclusion. |
|
3. |
References |
Some updated references have been inserted in the manuscript |
Reviewer 2
Round 2
Reviewer 2 Report
The authours have sufficiently improved this review. Though there could have been more discussion (and corresponding citations) on the design and characterization of nanovectors, I recognize that an exahustive treament of this subject is a formidable task. Therefore, I agree that the manuscript can be published in its current form